# Examining the Validity of ChatGPT in Identifying Relevant Nephrology Literature: Findings and Implications

**DOI:** 10.3390/jcm12175550

**Published:** 2023-08-25

**Authors:** Supawadee Suppadungsuk, Charat Thongprayoon, Pajaree Krisanapan, Supawit Tangpanithandee, Oscar Garcia Valencia, Jing Miao, Poemlarp Mekraksakit, Kianoush Kashani, Wisit Cheungpasitporn

**Affiliations:** 1Division of Nephrology and Hypertension, Department of Medicine, Mayo Clinic, Rochester, MN 55905, USA; s.suppadungsuk@hotmail.com (S.S.); charat.thongprayoon@gmail.com (C.T.); pajaree_fai@hotmail.com (P.K.); supawit_d@hotmail.com (S.T.); garciavalencia.oscar@mayo.edu (O.G.V.); miao.jing@mayo.edu (J.M.); mekraksakit.poemlarp@mayo.edu (P.M.); kashani.kianoush@mayo.edu (K.K.); 2Chakri Naruebodindra Medical Institute, Faculty of Medicine Ramathibodi Hospital, Mahidol University, Samut Prakan 10540, Thailand; 3Division of Nephrology, Thammasat University Hospital, Pathum Thani 12120, Thailand

**Keywords:** ChatGPT, nephrology literature, references, reliability, accuracy

## Abstract

Literature reviews are valuable for summarizing and evaluating the available evidence in various medical fields, including nephrology. However, identifying and exploring the potential sources requires focus and time devoted to literature searching for clinicians and researchers. ChatGPT is a novel artificial intelligence (AI) large language model (LLM) renowned for its exceptional ability to generate human-like responses across various tasks. However, whether ChatGPT can effectively assist medical professionals in identifying relevant literature is unclear. Therefore, this study aimed to assess the effectiveness of ChatGPT in identifying references to literature reviews in nephrology. We keyed the prompt “Please provide the references in Vancouver style and their links in recent literature on… name of the topic” into ChatGPT-3.5 (03/23 Version). We selected all the results provided by ChatGPT and assessed them for existence, relevance, and author/link correctness. We recorded each resource’s citations, authors, title, journal name, publication year, digital object identifier (DOI), and link. The relevance and correctness of each resource were verified by searching on Google Scholar. Of the total 610 references in the nephrology literature, only 378 (62%) of the references provided by ChatGPT existed, while 31% were fabricated, and 7% of citations were incomplete references. Notably, only 122 (20%) of references were authentic. Additionally, 256 (68%) of the links in the references were found to be incorrect, and the DOI was inaccurate in 206 (54%) of the references. Moreover, among those with a link provided, the link was correct in only 20% of cases, and 3% of the references were irrelevant. Notably, an analysis of specific topics in electrolyte, hemodialysis, and kidney stones found that >60% of the references were inaccurate or misleading, with less reliable authorship and links provided by ChatGPT. Based on our findings, the use of ChatGPT as a sole resource for identifying references to literature reviews in nephrology is not recommended. Future studies could explore ways to improve AI language models’ performance in identifying relevant nephrology literature.

## 1. Introduction

Current approaches to identifying references for literature reviews in medical research can be challenging and time-consuming. Researchers typically rely on manual searches of databases, such as PubMed or Scopus, using keywords and filters to retrieve relevant articles [1]. This process often involves iterative searches and screening of a large volume of articles, which can be overwhelming and prone to missing essential sources [2,3]. Moreover, the availability and accessibility of literature vary across different databases and journals. Researchers may encounter restricted access, which can be in the form of paywalls for which paid subscriptions must be purchased to access the full-text articles. The limiting ability to retrieve and include relevant references might lead to inadequate coverage of the literature reviews and potential bias in reference selection. Additionally, the process of identifying relevant references requires expertise and domain knowledge. Researchers must have a comprehensive knowledge, and many of them require the expertise of a dedicated librarian in their efforts to retrieve the needed literature. While subjective evaluation can introduce potential biases and inconsistencies in the selection of references, bias assessment tools are widely employed to mitigate such challenges, ensuring a systematic and objective approach to literature selection.

Artificial intelligence (AI)-assisted approaches, such as the model Chat Generative Pretrained Transformer (ChatGPT) [4], offer promising solutions to the challenges faced in traditional reference identification methods, thereby potentially enhancing efficiency and accuracy. ChatGPT, developed by OpenAI, is a cutting-edge natural language processing (NLP) model that leverages the power of AI to comprehend and generate human-like responses through text or voice interactions [5]. This sophisticated chatbot can potentially revolutionize various domains, particularly education in healthcare [6,7,8]. It offers enhanced search capabilities, discovering relevant references and information in language-based tasks [9,10,11]. The applications of ChatGPT extend beyond basic language-related tasks. It can improve academic writing by identifying and correcting grammar and spelling errors, enhancing the clarity of topics, and providing personalized learning experiences [10,12,13,14]. Additionally, ChatGPT proves valuable in data analysis, literature reviews, and manuscript writing, offering researchers a time-saving tool to streamline their work [15,16,17]. ChatGPT and similar AI models can swiftly search through extensive volumes of literature, retrieving relevant articles based on user queries or prompts. These models can comprehend and interpret natural language, enabling researchers to effectively communicate their search criteria and receive targeted results [18,19]. By evaluating the efficacy of ChatGPT in the context of literature reviews in nephrology, researchers can better understand its potential benefits and limitations [18,20,21,22]. This evaluation can guide the development of strategies to improve ChatGPT’s performance in identifying accurate and relevant references and addressing concerns related to data bias, information accuracy, and citation errors. Ultimately, integrating AI-assisted approaches like ChatGPT into the literature review process can save researchers valuable time, enhance the comprehensiveness of literature coverage, and contribute to evidence-based decision-making in nephrology and other medical fields.

One key advantage of AI models is their capacity to overcome the limitations of traditional approaches by providing access to a broader range of literature sources [22,23,24,25,26,27]. They can integrate data from multiple databases and journals, including those that may not be readily accessible or commonly searched [16,28]. This expanded scope increases the comprehensiveness of literature reviews, minimizing the risk of overlooking relevant references. Moreover, AI models can assist researchers in assessing the quality and relevance of retrieved articles. By analyzing various parameters such as citation counts, journal impact factors, author credentials, and content similarity, these models can provide additional insights and suggestions to aid in selecting appropriate references. However, the use of ChatGPT in medical research has potential drawbacks, as highlighted by previous studies [18,21,28,29,30]. Concerns have been raised about the data bias, inaccurate information, and citation errors associated with ChatGPT [12,31]. 

To better understand the benefits and limitations of ChatGPT in the context of literature reviews in nephrology, an evaluation of its efficacy is necessary. This assessment can guide the development of strategies to enhance ChatGPT’s performance in accurately identifying relevant references while addressing concerns related to data bias, information accuracy, and citation errors. Integrating AI-assisted approaches like ChatGPT into the literature review process can save researchers valuable time, improve the comprehensiveness of literature coverage, and contribute to evidence-based decision-making in nephrology and other medical fields.

Considering the undetermined effectiveness of ChatGPT in assisting medical professionals with identifying relevant literature in nephrology, this study aims to assess the effectiveness of ChatGPT in identifying references for literature reviews, specifically in the field of nephrology and its specific subdomains. By evaluating ChatGPT’s performance, researchers can gain insights into its potential benefits and limitations, contributing to improving AI-assisted tools for literature review processes.

## 2. Materials and Methods

### 2.1. Search Strategy and Criteria

To conduct this study, we utilized ChatGPT, an AI chatbot developed by OpenAI. Specifically, we employed the Generative Pre-trained Transformer model (GPT-3.5) within ChatGPT to search for topics in nephrology and specific subdomains. All of the areas in nephrology that our study selected were based on the critical aspects of nephrology, including (1) general nephrology, (2) glomerular disease, (3) hypertension, (4) acute kidney injury, (5) chronic kidney disease, (6) end-stage kidney disease, (7) electrolyte disorders, (8) acid-base disturbances, (9) kidney stones, (10) hemodialysis, (11) peritoneal dialysis, and (12) kidney transplantation.

The search prompts provided to ChatGPT requested references in the Vancouver style, a commonly used citation style in academic writing, along with their corresponding links. We generated the prompt “Please provide the references in Vancouver style and their links in recent literature on… name of the topic” to ChatGPT. We documented six key components for each identified reference, including (1) authors, (2) reference titles, (3) journal names, (4) publication years, (5) digital object identifiers (DOIs), and (6) reference links.

To verify the existence and accuracy of the reference citations, we employed multiple reliable sources, including PubMed, Google Scholar, and Web of Science. For each reference, we first used the provided DOI to search for its corresponding publication in PubMed, the widely recognized and trusted database for biomedical literature. If a reference was found in PubMed, it was considered existing and authentic. In cases where PubMed did not yield any results or when we encountered incomplete or missing DOIs, we used Google Scholar as an additional resource. Google Scholar is a comprehensive search engine that indexes various scholarly articles from multiple disciplines, including medical and non-medical literature, such as engineering, arts, humanities, and beyond. We searched using the reference titles, authors’ names, and other relevant information to locate the references and confirm their validity. Additionally, we utilized the Web of Science database, a renowned research platform covering multiple disciplines, to cross-reference the references obtained. An authentic reference was defined as a citation that existed and could be verified for accuracy. All six components, namely authors’ names, reference titles, journal names, publication years, DOIs, and reference links, had to be correct for a reference to be considered authentic.

By employing these three prominent databases—PubMed, Google Scholar, and Web of Science—we aimed to thoroughly assess the authenticity and accuracy of the references generated by ChatGPT. This approach allowed us to validate the existence and authenticity of the references within an academic context. It ensured that the citations were based on reliable and reputable sources in the field of nephrology. Our study only examined the references in English. For other language contexts, there was a limitation in evaluating the validation of the reference.

### 2.2. Study Outcomes

The primary objective of this study was to assess the validity of references generated by ChatGPT within an academic context. Validity encompassed the authenticity of the references, including authors’ names, topics, journal names, publication years, DOIs, and links, which had to be correct.

References were categorized as fabricated or non-existent if all citation elements were forged or non-existent, respectively. Existing references were deemed incorrect if at least one component was inaccurate. References with incomplete sets of six elements were classified as incomplete.

In addition to assessing the accuracy, we evaluated the frequency of incorrect components within each reference and across different nephrology subdomains.

### 2.3. Statistical Analysis

Descriptive statistics were employed to present the data in numbers and percentages. IBM SPSS Statistics version 26 was utilized for all statistical analyses. This software allowed us to summarize and analyze the data. The validity of references generated by ChatGPT was evaluated in terms of the completeness of references, fabrication, and authenticity, which were presented as percentages and numbers using descriptive statistics.

## 3. Results

A total of 610 references were provided from the ChatGPT search in 12 topics of specific fields of nephrology. Of the references given by ChatGPT, we found that 378 (62%) existed, while 192 (31%) were fabricated, and 40 (7%) were incomplete (Figure 1). The examples of incomplete, fabricated, and inaccurate references are demonstrated in Figure 2. Among the existing references, 60.3% of those provided by ChatGPT were relevant to the specific topic. Furthermore, 20% of the citations were identified as accurate, meeting all six of the components required for authenticity.

Figure 3 shows the percentage of fabricated references based on specific nephrology domains. The percentage of fabricated references ranged from 1% in acute kidney injury and general nephrology to 18% in the electrolyte domain.

When we analyzed the accuracy of reference components, among the six components, an inaccurate link was the most common, noted in 68% of references, followed by DOI (54%), journal (14%), year (10%), author (3%), and title (0.3%) (Figure 4).

Table 1 presents the results of the subgroup analysis in the specific fields of nephrology. The authenticity of references provided by ChatGPT was highest in the general nephrology area, with only 62% of references being considered authentic. In other fields, the percentage of authentic references fell below 50%. Notably, none of the references generated by ChatGPT about peritoneal dialysis were found to be authentic. 

## 4. Discussion

In recent years, the influence of AI has expanded across various aspects of human life, with ChatGPT emerging as a widely used and powerful tool. While ChatGPT has been helpful in healthcare education and research [11,32,33], concerns about the reliability and accuracy of data, particularly in nephrology, have arisen [34,35]. This study aims to evaluate the effectiveness of ChatGPT in identifying authentic references for literature reviews in the various fields of nephrology and to determine the accuracy of each component in nephrology and specific nephrology areas provided by ChatGPT.

Our findings demonstrate that, out of the 610 references generated by ChatGPT across various nephrology fields, one-third were fabricated. Most fabricated references (60%) were found in the fields of electrolyte disorders, kidney stones, and hemodialysis. It is noteworthy that only 20% of the citations were authentic. Among the fields studied, general nephrology exhibited the highest accuracy and reliability (62%) and the lowest rate of fabricated citations (4%) from ChatGPT. Furthermore, in the existing references provided by ChatGPT, the most common inaccuracies were related to incorrect links and DOIs.

Similar concerns have been raised in previous studies regarding the plausibility of references provided by ChatGPT [12,20,29,36,37]. Compared to earlier studies on medical content provided by ChatGPT [38,39], our findings demonstrate similarities in terms of fabricated references, although at different rates (31% vs. 47%) and with a higher percentage of authentic references (20% vs. 7%). Additionally, our study reveals a lower rate of incorrect authors and titles for existing references, with <5% of these components being incorrect, whereas previous studies reported rates exceeding 40% [38]. These differences may be attributed to our study’s larger sample size and the focus on nephrology fields in the input to ChatGPT. Furthermore, it is possible that the hallucination effect of ChatGPT, influenced by the e-data input or prompts to the ChatGPT program and potential updates in the algorithm, differs between studies. We also observed that, when ChatGPT provided the correct link, all six components of the references were authentic. Given this observation, we postulated that there might exist a potential correlation between the presence of a correct link and the authenticity of the associated reference. This hypothesis warrants further exploration in future studies to confirm any causal relationship. 

Future studies in AI and Large Language Models, like ChatGPT, have the potential for advancing the accuracy and reliability of generated references in various domains, including nephrology. The limitations identified in our study call for further research and development to address these issues and improve the performance of AI language models. One possible direction for future studies is to explore the use of more advanced versions of AI models, such as GPT 4.0, which may offer enhanced capabilities and improved accuracy in generating references [40]. Additionally, researchers can investigate the integration of additional data sources and databases beyond Google Scholar to verify the authenticity and reliability of references provided by AI models. Moreover, future studies should focus on developing techniques to evaluate the quality and relevance of generated references more comprehensively. This could involve analyzing the content of each reference, including abstracts and full-text articles, to ensure that the information aligns with the specified topic and meets the desired criteria for inclusion in literature reviews. Implementing NLP techniques and machine learning algorithms can aid in assessing the semantic relevance of references and identifying potential inaccuracies or fabrications.

Employing pre-trained models, specifically trained on high-quality, curated datasets of references from reputable sources to improve the reliability of references, can be trained in future studies. By training the AI models on reliable references, the generated outputs are more likely to be accurate and trustworthy. Researchers can also collaborate with domain experts and nephrology professionals to curate and validate reference databases specifically tailored to the field of nephrology. This domain-specific curation can help AI models generate more relevant and authentic references. Furthermore, future studies should aim to address the issue of incomplete and incorrect reference components. AI models can be trained to recognize and validate different components of references, such as authors, titles, DOIs, and links, to ensure that all elements are accurate and complete. Additionally, cross-referencing with multiple databases and implementing automated fact-checking algorithms can help identify and rectify inaccuracies in the generated references.

The implications for future advancements in AI and language models for nephrology are far-reaching [22]. The improved accuracy and reliability of generated references could significantly benefit researchers, clinicians, and educators. Literature reviews play a crucial role in evidence-based medicine and research, and reliable references are essential for making informed decisions and drawing accurate conclusions. By leveraging AI models effectively, researchers can save time and effort in the literature review process, allowing them to focus more on data analysis and interpretation. Moreover, AI language models like ChatGPT can serve as powerful educational tools in nephrology. They can assist in providing up-to-date and relevant references to students, trainees, and healthcare professionals, facilitating their learning and professional development [41]. AI models can offer quick access to a wide range of literature, helping users stay updated with the latest advancements, guidelines, and research findings in nephrology.

In addition to literature reviews, AI models can be further developed to assist in other nephrology research and practice aspects. For example, AI-powered systems can be trained to extract relevant information from large datasets, such as electronic health records and clinical trial data, to identify patterns, predict outcomes, and improve patient care [42]. AI algorithms can also aid in automating the identification and diagnosis of kidney diseases, the analysis medical imaging data, and the optimization of treatment plans. However, it is essential to approach the integration of AI and language models in nephrology with caution. The reliability and accuracy of AI-generated references should always be verified and cross-checked by human experts. AI should be seen as a valuable tool to support and enhance the work of researchers and healthcare professionals rather than a substitute for their expertise and critical thinking [43,44]. Ethical considerations, data privacy, and transparency in AI algorithms should be a priority.

Several limitations should be acknowledged in our study. Firstly, we used the free version of ChatGPT, GPT 3.5, which may not possess the same level of accuracy and reliability as the paid version, GPT 4.0. This difference in performance could potentially impact the output results. Secondly, our study focused on the citation components obtained through prompts to ChatGPT without thoroughly examining each study’s abstract or detailed content. Finally, we did not utilize the extension mode of ChatGPT to assess the latest updates of references. Our study employed the free version of ChatGPT (GPT-3.5), which has limitations since GPT-3.5 has been trained and comprehends data updates through September 2021 [45]. This may have resulted in missing out on the benefits of improved capabilities and potentially more accurate generated results.

## 5. Conclusions

Our findings support the notion that relying solely on references provided by ChatGPT, without considering the potential for artificial hallucination, poses risks of unreliable and inaccurate references. The use of ChatGPT as the sole resource for identifying references in nephrology literature reviews is not recommended. Future studies should explore ways to improve the performance of AI language models in identifying relevant literature in nephrology.

## Figures and Tables

**Figure 1 jcm-12-05550-f001:**
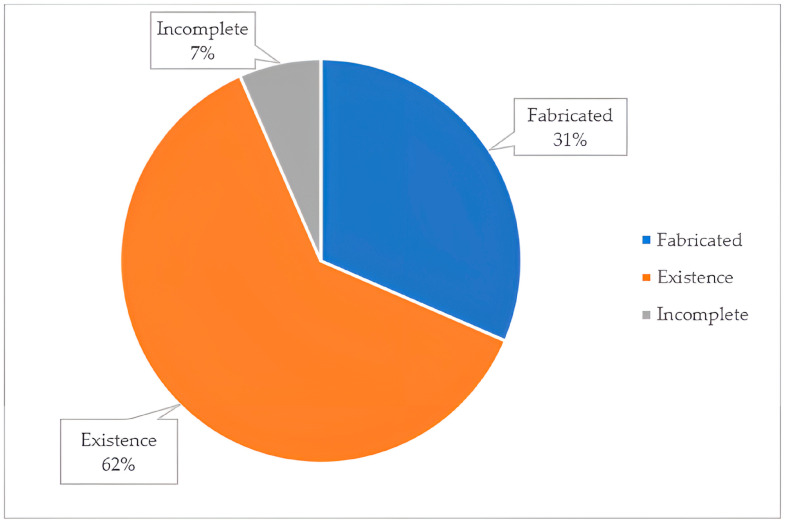
The reliability of references provided by ChatGPT.

**Figure 2 jcm-12-05550-f002:**
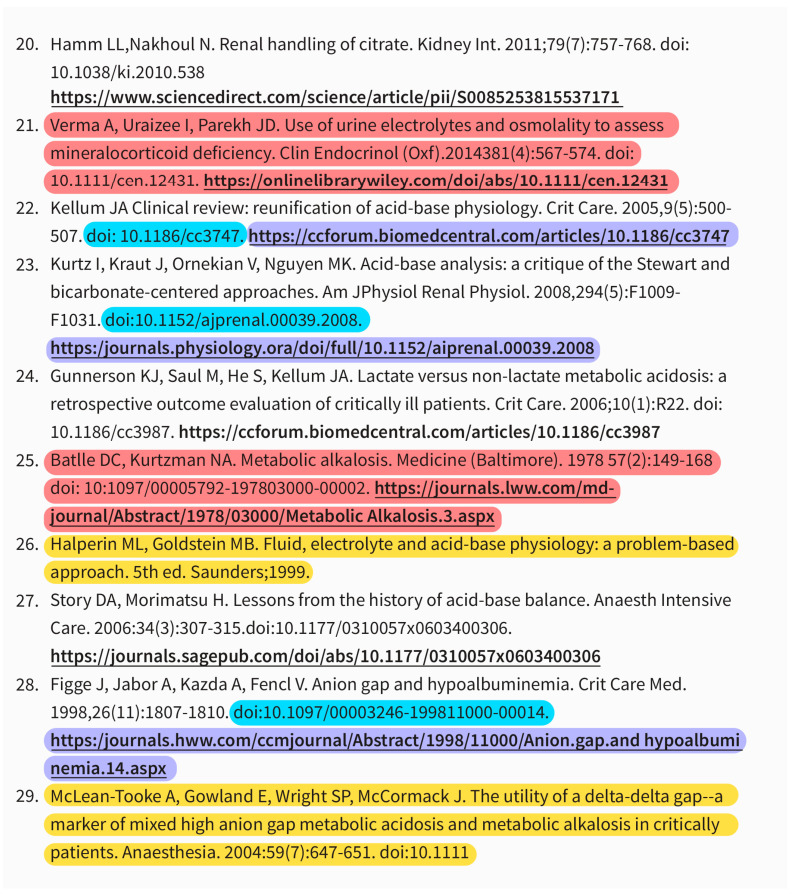
The example of incomplete, fabricated, and incorrect references provided by ChatGPT. The red color referred to the fabricated references, for which all citation components were inaccurate, including authors, title, journal name, year of publication, digital object identifiers (DOI), and link. The blue highlight was incorrect DOI number. The purple was inaccurate link. The yellow highlighted represented incomplete references, which did not have all six components. The underline was the hyperlink provided by ChatGPT.

**Figure 3 jcm-12-05550-f003:**
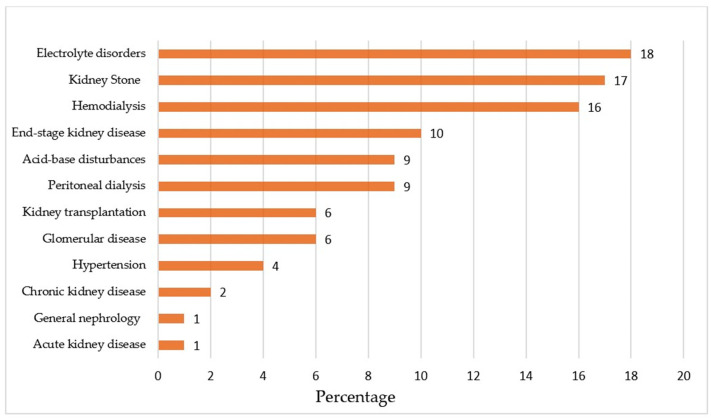
The percentage of fabricated references in the specific fields of nephrology.

**Figure 4 jcm-12-05550-f004:**
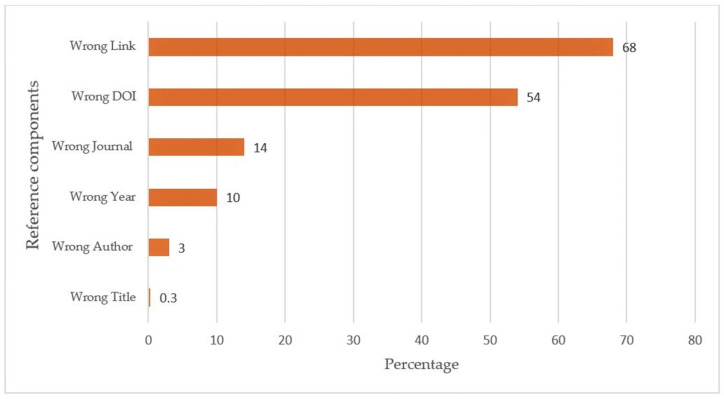
The percentage of inaccurate individual reference components generated by ChatGPT. DOI: digital object identifier (DOI).

**Table 1 jcm-12-05550-t001:** Evaluation in specific fields of nephrology.

	Total	Incomplete Reference (%)	Complete Reference		
			Fabricated (%)	Existence with All Correct (%)	Existence with Partial Correct (%)
Overall	610	40 (7)	192 (31)	122 (20)	256 (42)
Subgroups					
Acute kidney disease	51	2 (4)	2 (4)	12 (24)	35 (68)
General Nephrology	50	4 (8)	2 (4)	31 (62)	13 (26)
Glomerular disease	50	1 (2)	11 (22)	5 (10)	33 (66)
Chronic kidney disease	52	4 (8)	4 (8)	13 (25)	31 (59)
Hemodialysis	51	4 (8)	31 (60)	2 (4)	14 (28)
Electrolyte disorders	51	0 (0)	35 (68)	4 (8)	12 (24)
Acid-base disturbances	50	8 (16)	18 (36)	8 (16)	16 (32)
End-stage kidney disease	55	7 (13)	20 (36)	8 (15)	20 (36)
Hypertension	50	5 (10)	7 (14)	22 (44)	16 (32)
Kidney Stone	50	2 (4)	33 (66)	3 (6)	12 (24)
Kidney transplantation	50	2 (4)	11 (22)	14 (28)	23 (46)
Peritoneal dialysis	50	1 (2)	18 (36)	0 (0)	31 (62)

## Data Availability

Data supporting this study are available in the original publication, reports, and preprints that were cited in the reference citation.

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
