# Peer review of "Examining the Validity of ChatGPT in Identifying Relevant Nephrology Literature: Findings and Implications"

_jcm, 2023, doi:10.3390/jcm12175550_

Round 1

Reviewer 1 Report

The article untitle Examining the Validity of ChatGPT in Identifying Relevant Nephrology Literature: Findings and Implications is a novel article with the aims to evaluate the effectiveness and accuracy of ChatGPT in identifying references for literature reviews in nephrology and specific nephrology areas. Authors have demostrated that out of 610 references generated by ChatGPT across various nephrology fields, one-third were fabricated. The interesting findings allow authors conclude that relying solely on references provided by ChatGPT without considering the potential for artificial hallucination poses risks of unre liable and inaccurate references. The use of ChatGPT as the sole resource for identifying references in nephrology literature reviews is not recommended.

Line 170: change Figure 3 Demonstration of the percentage... by Figure 3 shows the percentage...

Line 174: Figure 3. The percentage of fabricated references in the specific fields of Nephrology. (period)

Author Response

Response to Reviewer#1

Reviewer 1

Comments and Suggestions for Authors

The article entitled Examining the Validity of ChatGPT in Identifying Relevant Nephrology Literature: Findings and Implications is a novel article with the aims to evaluate the effectiveness and accuracy of ChatGPT in identifying references for literature reviews in nephrology and specific nephrology areas. Authors have demonstrated that out of 610 references generated by ChatGPT across various nephrology fields, one-third were fabricated. The interesting findings allow authors conclude that relying solely on references provided by ChatGPT without considering the potential for artificial hallucination poses risks of unreliable and inaccurate references. The use of ChatGPT as the sole resource for identifying references in nephrology literature reviews is not recommended.

Response: Thank you for your thoughtful review and constructive feedback on our manuscript titled "Examining the Validity of ChatGPT in Identifying Relevant Nephrology Literature: Findings and Implications." We are pleased that you found our research novel and its implications valuable. We are committed to ensuring our work meets the highest standards of clarity and quality, and we have thoroughly addressed the comments you provided.

Comment #1

Comments on the Quality of English Language

Response:

We sincerely appreciate your feedback and consideration of our manuscript. Ensuring clarity and precision in our work is of paramount importance, and your insights have been invaluable in this endeavor. We have carefully revised the areas you highlighted to improve the quality of the English language in our manuscript.

  1. Line 170 - Original Sentence: Figure 3 Demonstration of the percentage of fabricated references based on specific nephrology domains.

Revised Sentence: Figure 3 shows the percentage of fabricated references based on specific nephrology domains.

  1. Line 174 - Original Sentence: The percentage of fabricated references in the specific fields of Nephrology.

Revised Sentence: The percentage of fabricated references in specific nephrology fields.

We have thoroughly reviewed the entire manuscript and made further adjustments as needed to ensure the quality of English language throughout. We hope that these revisions address your concerns, and we are open to any further suggestions or clarifications.

Once again, thank you for your time and expertise in reviewing our work. We believe that your feedback has significantly enhanced the quality and clarity of our manuscript, and we are grateful for your contribution to this process.

Reviewer 2 Report

The authors have dealt with an extremely topical and original issue.  In fact, the use of artificial intelligence (AI) in scientific research is highly debated.

The english language is fluent, the paper has a clear structure and the study is well designed. The presented findings, along with figures and tables,  well describe the existing pitfalls of the use of AI in medical research. Although it could be a helpful tool in the future, the authors correctly call for cautiousness and currently discourage the use of AI as the sole resource for identifying references. Importantly, they support their conclusions with robust data.

Author Response

Response to Reviewer#2

Comment #1

The authors have dealt with an extremely topical and original issue.  In fact, the use of artificial intelligence (AI) in scientific research is highly debated.

The English language is fluent, the paper has a clear structure and the study is well designed. The presented findings, along with figures and tables, well describe the existing pitfalls of the use of AI in medical research. Although it could be a helpful tool in the future, the authors correctly call for cautiousness and currently discourage the use of AI as the sole resource for identifying references. Importantly, they support their conclusions with robust data.

Response: We are sincerely grateful for taking your precious time to review our manuscript. Your comprehensive feedback reinforces our motivation to tackle such a pertinent subject in the medical research field. We concur that while AI holds transformative potential, its prudent and judicious application is crucial. Your emphasis on the robustness of our data and the clarity of our figures and tables resonates with our commitment to produce high-quality research.

In light of your feedback, we've revisited the manuscript to ensure that our cautionary stance on AI's current capabilities, in the context of our research, is clear and appropriately supported by the data presented. Your insight, coupled with the depth of your review, has been instrumental in enhancing the quality of our manuscript. We remain open to any further suggestions or feedback you might have to refine our work even further.

Once more, thank you for your invaluable contribution to the enhancement of our research through your review.

Thank you for your time and consideration.  We greatly appreciated the reviewer's and editor's time and comments to improve our manuscript. The manuscript has been improved considerably by the suggested revisions.

Reviewer 3 Report

Thanks, dear colleagues, for sharing this wonderful research paper!

Here are comments to further empower your manuscript:

Abstract:

Line 14: "...(AI) language model..." is better called "...(AI) Large Language Model (LLM)...", and may use this LLM in the manuscript as ChatGPT and other AI Chatbots are referred to in such literature

Line 18: "ChatGPT (03/23 Version)." is this ChatGPT-3.5 ?

Introduction:

Line 35: consider changing to "Current approaches..." instead of "Traditional approaches..."

Line 41: "Researchers may encounter paywalls...": what do you mean with paywalls? is it subscription-based publications?

Line 45: "Researchers must have a comprehensive ....": may add here that many of them require the expertise of a dedicated librarian in their efforts to retrieve the needed literature

Line 47: "This subjective evaluation can introduce bias and inconsistencies in the selection of references.": but that is why there is bias assessment?

Line 49: Please add full spelling before the first ChatGPT appearance in the paper

Line 50:  should be: "thereby potentially enhancing efficiency and accuracy."

Line 65: As mentioning ChatGPT and nephrology in the literature, may consider adding this reference: PMID: 37278392

Line 74: "... to a broader range of literature sources.": Please add Ref

Materials and Methods:

Lines 101-107: Please elaborate how were the nephrology topics you used were selected? Was it your group's expert opinion or by importance or recurrence to the field of Nephrology? 

Line 108: Please state the exact prompt and when was it used on ChatGPT? Also, how was the prompt developed/engineered?

Line 120: "...various scholarly articles from multiple disciplines" do you mean medical and non-medical literature?

Line 132: "...based on reliable and reputable sources in the field of nephrology." does the search strategy you used and the databases you verified cover other languages than English? If not, this should be clarified in the Limitations

Line 135: "authenticity and accuracy": please define each, if these are different terms

Statistical Analysis:
"This software allowed us to summarize and analyze the data, providing insights into the validity of the references generated by ChatGPT and the frequency of incorrect components within different subdomains of nephrology.": Please elaborate on which tests were utilized to achieve this.

Results:

Line 153: please add number before % in "7% were incomplete"

Figure 2: is there any copyright issue of using ChatGPT logo in your figure?

Table 1 setup is confusing: I had to calculate the numbers to figure out that Complete reference column was then divided into 3 parallel columns (Fabricated, Existence with all correct, Existence with partial correct) so consider either splitting/dividing the table more clearly or making it into 2 Tables to avoid confusion. Also, please explain that () is (%).

 Discussion:

Line 193: "and accuracy of data, particularly in nephrology, have arisen." I suggest adding ref, such as PMID: 36754723 PMID: 37229893

 Line 193: Please elaborate here on the terms "effectiveness and accuracy of ChatGPT in identifying references"

Lines 196-202: Please contrast with the literature and add Refs to this paragraph

Line 212: please rephrase from " data input to the program" to become "e data input, or prompts, to the ChatGPT program"

Line 215: " It is possible that all references 215 generated by ChatGPT with correct links were authentic." Was this a finding in your result? Please explain

Line 217: "language models like ChatGPT" is better phrased as"Large Language Models, like ChatGPT,"

Line 217: "hold immense potential for" This is a very strong statement for something that is still evolving and under research

Line 232: "Future studies..." You already started the preceding paragraph with such words, so I suggest rephrasing one of them

Line 244: "The implications of future advancements in AI and language models for nephrology are far-reaching." Suggested Ref: PMID: 37278392

Line 251: "They can assist in providing up-to-date and relevant references to students, trainees, and healthcare professionals, facilitating their learning and professional development": Suggested Ref: DOI: 10.7759/cureus.43036

Line 265: "...enhance the work of researchers and healthcare professionals rather than a substitute for...": consider adding Ref like: PMID: 37444647, PMID: 37162073

Line 274: Please rephrase for clarity "...as GPT-3.5 has limitations in updating data before 2022."

Some commas are missing to make text better readable

Author Response

Response to Reviewer#3

Thanks, dear colleagues, for sharing this wonderful research paper!

Here are comments to further empower your manuscript:

Response: We deeply value your constructive and positive feedback on our manuscript. Your recommendations have indeed been instrumental in making our paper more accurate and thorough. In response, we have revised the sentence as follows:

Abstract:

  1. Original sentence: ChatGPT is a novel artificial intelligence (AI) language model renowned for its exceptional ability to generate human-like responses across various tasks.

Response: Your suggestion to elucidate the type of AI model by indicating "Large Language Model (LLM)" provides greater clarity to the readers unfamiliar with the terminology. We appreciate this input and have updated the sentence accordingly.

Revised Sentence: ChatGPT is a novel artificial intelligence (AI) Large Language Model (LLM) renowned for its exceptional ability to generate human-like responses across various tasks.

  1. Line 18: "ChatGPT (03/23 Version)." is this ChatGPT-3.5?

Original Sentence: We keyed the prompt "Please provide the references in Vancouver style and their links in recent literature on... name of the topic" into ChatGPT (03/23 Version).

Response: We appreciate your attention to detail. Your inquiry regarding the specific version of ChatGPT utilized in our research has prompted us to provide a more explicit version clarification, ensuring readers have a clear understanding of the tools we employed. The manuscript has been revised to reflect the use of ChatGPT-3.5 (03/23 Version) as suggested.

Revised Sentence: We keyed the prompt "Please provide the references in Vancouver style and their links in recent literature on... name of the topic" into ChatGPT-3.5 (03/23 Version).

Introduction:

  1. Line 35: Original Sentence: Traditional approaches to identifying references for literature reviews in medical research can be challenging and time-consuming.

Response: Thank you for suggesting a shift in terminology from "Traditional" to "Current". This subtle change indeed provides a more accurate representation of the present-day scenario, indicating the ongoing challenges faced by researchers in the field.

Revised Sentence: Current approaches to identifying references for literature reviews in medical research can be challenging and time-consuming.

  1. Original Sentence: Researchers may encounter paywalls or restricted access to full-text articles, limiting their ability to retrieve and include relevant references in their literature reviews.

Response: Your revision insightfully dissects the hindrances researchers face and elaborates on the consequences of these barriers. The modified structure offers a clearer understanding of the cascade of challenges that arise from restricted access. We believe this enhancement paints a fuller picture for our readers.

Revised Sentence: Researchers may encounter restricted access, which can be in the form of a paywall that must be purchased or paid subscription to access the full-text articles. The limiting ability to retrieve and include relevant references might lead to inadequate coverage of the literature reviews and potential bias in reference selection.

  1. Original Sentence: Researchers must have a comprehensive understanding of the subject area and be able to assess the relevance and quality of each article.

Response: This revision highlights the depth of expertise and the collaborative effort often needed in literature retrieval. By introducing the role of the dedicated librarian, we emphasize the complexity and meticulous nature of the process. This valuable addition paints a realistic picture of the current state of literature review practices.

Revised Sentence: Researchers must have a comprehensive that many of them require the expertise of a dedicated librarian in their efforts to retrieve the needed literature.

  1. Line 47: "This subjective evaluation can introduce bias and inconsistencies in the selection of references.": but that is why there is bias assessment?

Response:  Indeed, you are right. Bias assessments are integral to counteract the potential inconsistencies and biases introduced by subjective evaluations. The primary objective of bias assessment tools is to ensure the integrity and validity of the research by systematically identifying and mitigating biases. Our intention was to underline the inherent challenges faced during the selection of references, particularly when the process is heavily reliant on individual judgment. However, we recognize the importance of elaborating on the safeguarding role of bias assessment tools in this context. To provide a more balanced view, we have revised the sentence as follows:

Revised Sentence: "While subjective evaluation can introduce potential biases and inconsistencies in the selection of references, bias assessment tools are widely employed to mitigate such challenges, ensuring a systematic and objective approach to literature selection."

  1. Original Sentence: Artificial intelligence (AI)-assisted approaches, such as ChatGPT

Response: We acknowledge the need for precision and clarity, especially when introducing terminologies that might be unfamiliar to some readers. Your suggestion provides an opportunity to present a more comprehensive introduction to ChatGPT.

Revised Sentence: Artificial intelligence (AI)-assisted approaches, such as model Chat Generative Pretrained Transformer (ChatGPT)

  1. Original Sentence: …offer promising solutions to the challenges faced in traditional reference identification methods, thereby enhancing efficiency and accuracy. 

Response: We concur with your observation. By introducing "potentially", we not only convey the prospective advantages of AI-assisted methods but also acknowledge that the realization of these benefits may vary based on specific contexts and implementations.

Revised Sentence:  …offer promising solutions to the challenges faced in traditional reference identification methods, thereby potentially enhancing efficiency and accuracy."

  1. Original Sentences: ChatGPT in the context of literature reviews in nephrology, researchers can better understand its potential benefits and limitations [18,20,21].

Response: We acknowledge your observation and have made the necessary adjustment.

Revised Sentence: ChatGPT in the context of literature reviews in nephrology, researchers can better understand its potential benefits and limitations [18,20-22].

  1. Original Sentence: One key advantage of AI models is their capacity to overcome the limitations of traditional approaches by providing access to a broader range of literature sources.

Response: We acknowledge the importance of substantiating our claims with appropriate references. Your suggestion to cite specific sources helps strengthen our assertion about the benefits of AI models.

Revised Sentence: One key advantage of AI models is their capacity to overcome the limitations of traditional approaches by providing access to a broader range of literature sources [22-27].

Materials and Methods:

  1. Please elaborate how were the nephrology topics you used were selected? Was it your group's expert opinion or by importance or recurrence to the field of Nephrology?

Response: We recognize the importance of elucidating the criteria used for topic selection in our study. Our decision to focus on the enumerated subdomains of nephrology was based on their significance and recurrence within the field, informed by both the collective expertise of our team and the broader consensus within the nephrology community about these pivotal areas.

Original Sentence: Specifically, we employed the Generative Pre-trained Transformer model (GPT-3.5) within ChatGPT to search for topics in nephrology and specific subdomains, including 1) general nephrology, 2) glomerular disease, 3) hypertension, 4) acute kidney injury, 5) chronic kidney disease, 6) end-stage kidney disease, 7) electrolyte disorders, 8) acid-base disturbances, 9) kidney stones, 10) hemodialysis, 11) peritoneal dialysis, and 12) kidney transplantation.

Revised Sentence: Specifically, we employed the Generative Pre-trained Transformer model (GPT-3.5) within ChatGPT to search for topics in nephrology and specific subdomains. All of the areas in nephrology that our study selected based on the critical aspects of nephrology, including, including 1) general nephrology, 2) glomerular disease, 3) hypertension, 4) acute kidney injury, 5) chronic kidney disease, 6) end-stage kidney disease, 7) electrolyte disorders, 8) acid-base disturbances, 9) kidney stones, 10) hemodialysis, 11) peritoneal dialysis, and 12) kidney transplantation.

We aimed to ensure our study spanned a comprehensive spectrum of nephrology, with a view to critically assessing ChatGPT’s efficacy in identifying pertinent literature. This would subsequently inform its potential applicability for future research endeavors in the field.

  1. Please state the exact prompt and when was it used on ChatGPT? Also, how was the prompt developed/engineered?

Response: We concur that an explicit elucidation of our prompt's design and its usage timeline provides clarity and transparency. The prompt employed with ChatGPT was meticulously crafted by our team with the intent to ensure that ChatGPT would understand our request clearly and provide the desired type of references. The development of this prompt was a deliberate process. We aimed to construct a clear, straightforward command that would effectively leverage the capabilities of ChatGPT, minimizing the chances of receiving irrelevant or extraneous outputs. We agree with the reviewer to explain how the command was created and display the precise prompt. The exact prompt that we used on ChatGPT was generated by our team. We keyed the prompt "Please provide the references in Vancouver style and their links in recent literature on... name of the topic"

Original Sentence: The search prompts provided to ChatGPT requested references in the Vancouver style, a commonly used citation style in academic writing, along with their corresponding links.  

Revised Sentence: The search prompts provided to ChatGPT requested references in the Vancouver style, a commonly used citation style in academic writing, along with their corresponding links.  We generated the prompt "Please provide the references in Vancouver style and their links in recent literature on... name of the topic" to ChatGPT.

  1. "...various scholarly articles from multiple disciplines" do you mean medical and non-medical literature?

Response: We acknowledge the need for clearer delineation in our phrasing and wholeheartedly agree with your observation. Your input illuminates the importance of specificity, ensuring that our readers are not left to interpret generalized statements. We revised the sentence as sentence below:

Revised Sentence: “Google Scholar is a comprehensive search engine that indexes various scholarly articles from multiple disciplines, including medical and non-medical literature, such as engineering, arts, humanities, and beyond.”

  1. "...based on reliable and reputable sources in the field of nephrology." does the search strategy you used and the databases you verified cover other languages than English? If not, this should be clarified in the Limitations

Response: We genuinely appreciate your insight on this matter. Upon reflection, we recognize that the language parameter of our search strategy was indeed confined to English. This limitation could potentially restrict the comprehensiveness of our review and omit valuable contributions from non-English sources. We totally agree with the reviewer that should be mentioned as the limitation. We have additionally added the following text in revised manuscript as suggested, “Our study examined the references only in English. For other language contexts, there was a limitation in evaluating the validation of the reference.”

  1. "authenticity and accuracy": please define each, if these are different terms

Response: We are grateful for your remark on the term authenticity and accuracy that we used. In terms of authenticity and accuracy in our context is the same meaning that the authentic reference was defined as all components of references must be correct. We have revised the sentence to improve clarity as follow:

“Validity encompassed the authenticity of the references, in which all components of the references, including authors' names, topics, journal names, publication years, DOIs, and links, must be correct.”

By this revised definition, we hope to emphasize the genuine and credible origin of a source (authenticity) while also ensuring that the information presented is correct and free from errors (accuracy).

Statistical Analysis:

  1. "This software allowed us to summarize and analyze the data, providing insights into the validity of the references generated by ChatGPT and the frequency of incorrect components within different subdomains of nephrology.": Please elaborate on which tests were utilized to achieve this.

Response: We acknowledge the importance of elaborating on the specific tests utilized in our study. In response to your suggestion, we have provided further clarity regarding our methodological approach:

Revised Statement: “IBM SPSS Statistics version 26 was utilized for all statistical analyses. This software allowed us to summarize and analyze the data. The validity of reference generated by ChatGPT was evaluated into the completeness of references, fabrication, and authenticity was presented as percentages and numbers using descriptive statistics.”

Results:

  1. Original Sentence: …, we found that 378 (62%) existed, while 192 (31%) were fabricated, and 7% were incomplete (Figure 1).

Response: We wholeheartedly concur with your recommendation to provide absolute numbers in conjunction with percentages. This approach offers a comprehensive and clearer representation of our findings. Our primary aim is to present our data with clarity and ensure the reader can easily grasp the outcomes of our study.

Revised Sentence: …, we found that 378 (62%) existed, while 192 (31%) were fabricated, and 40 (7%) were incomplete (Figure 1).

  1. Figure 2: is there any copyright issue of using ChatGPT logo in your figure?

Response: We sincerely appreciate your thorough review and insightful feedback on our work. Your attention to detail is instrumental in enhancing the quality of our research. We have carefully considered your concerns regarding the use of the ChatGPT logo in our figure and have taken prompt action to address the copyright concerns.  Upon reevaluation of the figure in question, we have diligently removed the ChatGPT logo to ensure alignment with copyright regulations and to respect OpenAI's guidelines. We understand the significance of safeguarding intellectual property rights and adhering to fair usage policies.

  1. Table 1 setup is confusing: I had to calculate the numbers to figure out that Complete reference column was then divided into 3 parallel columns (Fabricated, Existence with all correct, Existence with partial correct) so consider either splitting/dividing the table more clearly or making it into 2 Tables to avoid confusion. Also, please explain that () is (%).

Response: We agree with the reviewer that our design of Table 1. could be confusing. Your observation regarding the potential confusion arising from Table 1's design is genuinely appreciated. We recognize that in its current state, it may not be the most straightforward presentation for the reader, and we apologize for any inadvertent confusion caused. Upon reflecting on your feedback, we have taken action to revise Table 1. We have opted to restructure it to provide a more intuitive and clearer presentation of the data. We've chosen to categorize references into two main categories: "Incomplete" and "Complete" references. Subsequently, the "Complete" references are further subclassified into three distinct groups: "Fabricated," "Existence with all correct," and "Existence with partial correct."

To further ensure clarity, we have also included a footnote at the bottom of the table elucidating that the values presented within parentheses represent percentages. We trust that these modifications will rectify the initial confusion and present the data in a more organized and easily comprehensible manner.

Discussion:

  1. Line 193: Original Sentence: While ChatGPT has been helpful in healthcare education and research [11,32,33], concerns about the reliability and accuracy of data, particularly in nephrology, have arisen.

Response: We value your suggestion to include references [34,35] to support the claim about the reliability and accuracy concerns related to ChatGPT in the field of nephrology. This addition strengthens the statement and provides readers with the sources to understand the context better.

As such, we wholeheartedly accept your revision and have updated Line 193 as per your recommendation. This change not only makes the statement more evidence-based but also allows our readers to dive deeper into the specific concerns raised in nephrology regarding the utilization of ChatGPT.

Revised Sentence: While ChatGPT has been helpful in healthcare education and research [11,32,33], concerns about the reliability and accuracy of data, particularly in nephrology, have arisen [34,35].

  1. Please elaborate here on the terms "effectiveness and accuracy of ChatGPT in identifying references"

Response: We appreciate the need for clear definitions to enhance the reader's understanding. We are grateful for your mention regarding the term of effectiveness and accuracy outcome of this study. We have revised the sentence to improve clarity as follows: 

“This study aims to evaluate the effectiveness of ChatGPT in identifying authentic references for literature reviews in various fields of nephrology and determine the accuracy of each component in nephrology and specific nephrology area provided by ChatGPT.”

  1. Lines 196-202: Please contrast with the literature and add Refs to this paragraph

Response: We wholeheartedly agree that drawing parallels with the existing literature provides a comprehensive understanding and puts our results in context. However, it's worth noting that our work is pioneering in its examination of the validity of references specifically within the nephrology domain using ChatGPT. Therefore, direct comparisons within the same field are challenging. However, we've attempted to bridge this gap by contrasting our findings with other general medical studies involving ChatGPT. We have crafted the following paragraph to incorporate these comparisons:

“Similar concerns have been raised in previous studies regarding the plausibility of references provided by ChatGPT [12,20,29,36,37]. Compared to earlier studies on medical content provided by ChatGPT [38,39], our findings demonstrate similarities in terms of fabricated references, although at different rates (31% vs. 47%) and a higher percentage of authentic references (20% vs. 7%). Additionally, our study reveals a lower rate of incorrect authors and titles for existing references, with <5% of these components being incorrect, whereas previous studies reported rates exceeding 40% [38]. These differences may be attributed to our study's larger sample size and the focus on nephrology fields in the input to ChatGPT.”

We hope this paragraph offers a comprehensive contrast with available literature, elucidating the novelty and importance of our work.

  1. Original Sentence: Furthermore, it is possible that the hallucination effect of ChatGPT, influenced by the data input to the program and potential updates in the algorithm, differs between studies.

Response: We acknowledge your insightful suggestion to refine our statement, ensuring enhanced precision and clarity. The inclusion of terms like "e-data input" and "prompts to the ChatGPT program" indeed provide a more explicit understanding of the sources influencing ChatGPT's outputs. Upon reflection, we concur with your recommendation and have amended our sentence as follows:

Revised Sentence: Furthermore, it is possible that the hallucination effect of ChatGPT, influenced by the e-data input or prompts to the ChatGPT program and potential updates in the algorithm, differs between studies.

  1. Line 215: " It is possible that all references 215 generated by ChatGPT with correct links were authentic." Was this a finding in your result? Please explain

Response: We appreciate your thorough comment regarding the finding result. The data of our study show result that the mostly correct links in the references provided by ChatGPT were authentic. Upon revisiting our results and data analysis, we observed a trend wherein the references generated by ChatGPT that had accurate links predominantly turned out to be authentic. This observation formed the basis for our statement in Line 215. To ensure clarity and transparency, we have further revised, elaborated on this finding in our revised manuscript as follows:

“Given this observation, we postulated that there might exist a potential correlation be-tween the presence of a correct link and the authenticity of the associated reference. This hypothesis warrants further exploration in future studies to confirm any causal relation-ship. Although a correlation was apparent, further investigation is required to establish this association. It is possible that all references generated by ChatGPT with correct links were authentic.”

Our intention is to highlight this trend while also noting its preliminary nature, thereby prompting future studies to delve deeper into this potential correlation.

  1. Original Sentence: Future studies in AI and language models like ChatGPT hold immense potential for advancing the accuracy and reliability of generated references in various domains, including nephrology.

Response: Your suggestion is well-taken. Referring to ChatGPT specifically as a "Large Language Model" adds clarity and specificity to our statement, ensuring that readers unfamiliar with ChatGPT gain a better understanding of its nature and capabilities. The revised sentence, as you suggested, is now incorporated into our manuscript:

Revised Sentence: Future studies in AI and Large Language Models, like ChatGPT, hold immense potential for advancing the accuracy and reliability of generated references in various domains, including nephrology.  

  1. "hold immense potential for" This is a very strong statement for something that is still evolving and under research

Response: We appreciate for your careful consideration. We agree and thus have revised the sentence as follow:

“Future studies in AI and Large Language Models, like ChatGPT, has the potential for advancing the accuracy and reliability of generated references in various domains, including nephrology.”

  1. Original Sentence: Future studies can explore using pre-trained models specifically trained on high-quality, curated datasets of references from reputable sources to improve the reliability of references.

Response: We acknowledge the value in ensuring our manuscript remains clear and easily understood. We've reviewed your suggested revision and have made a further slight adjustment to enhance the flow of the sentence:

Revised Sentence: Employing pre-trained models specifically trained on high-quality, curated datasets of references from reputable sources to improve the reliability of references can be trained in future studies.

  1. Original Sentence: The implications of future advancements in AI and language models for nephrology are far-reaching.

Response: Your point about grounding our assertions in the existing literature is well-taken. Including reference [22] adds valuable context and credibility to our statement. We have incorporated the reference as suggested:

Revised Sentence: The implications of future advancements in AI and language models for nephrology are far-reaching [22].

  1. Original Sentence: They can assist in providing up-to-date and relevant references to students, trainees, and healthcare professionals, facilitating their learning and professional development.

Response: In accordance with your guidance, we have added the recommended reference to our revised sentence:

Revised Sentence: They can assist in providing up-to-date and relevant references to students, trainees, and healthcare professionals, facilitating their learning and professional development [41].

  1. Original Sentence: AI should be seen as a valuable tool to support and enhance the work of researchers and healthcare professionals rather than a substitute for their expertise and critical thinking.

Response: In alignment with your valuable feedback, we have revised our sentence to incorporate the suggested references:

Revised Sentence: AI should be seen as a valuable tool to support and enhance the work of researchers and healthcare professionals rather than a substitute for their expertise and critical thinking [43,44].

  1. Original Sentence: Finally, we did not utilize the extension mode of ChatGPT to assess the latest updates of references, as GPT-3.5 has limitations in updating data before 2022.

Response: To provide our readers with a clearer understanding of the limitations of our data source, we have adopted your suggested revision:

Revised Sentence: Finally, we did not utilize the extension mode of ChatGPT to assess the latest updates of references. Our study employed the free version of ChatGPT (GPT-3.5), which has limitations since GPT-3.5 has been trained and comprehends data updates through September 2021 [45].

Your feedback consistently aids us in enhancing the coherence and transparency of our manuscript. We remain grateful for your expert guidance.

Comments on the Quality of English Language

Some commas are missing to make text better readable

 Response: To address this, we have conducted a thorough review of the entire manuscript to identify and rectify instances where commas might be missing or punctuation could be improved. Our primary objective is to ensure that the document maintains the highest standards of readability and coherence.

We appreciate your diligence in pointing out this oversight. Ensuring clear communication of our findings and analysis is of paramount importance, and your feedback is instrumental in helping us achieve this goal.

Thank you for your time and consideration.  We greatly appreciated the reviewer's and editor's time and comments to improve our manuscript. The manuscript has been improved considerably by the suggested revisions.

Round 2

Reviewer 3 Report

Thanks for the revision and replies.

The revised paper is well-constructed and informative, I wish all the best to our colleagues!

Minor typos

Line 214: "...specific nephrology areas provide by ChatGPT.": provided

Line 231: e-data?